# Isolation of Bacterial and Fungal Microbiota Associated with *Hermetia illucens* Larvae Reveals Novel Insights into Entomopathogenicity

**DOI:** 10.3390/microorganisms10020319

**Published:** 2022-01-29

**Authors:** Patrick Klüber, Stephanie Müller, Jonas Schmidt, Holger Zorn, Martin Rühl

**Affiliations:** 1Branch for Bioresources, Fraunhofer Institute for Molecular Biology and Applied Ecology (IME), 35392 Giessen, Germany; patrick.klueber@ime.fraunhofer.de (P.K.); stephanie.mueller@lc.chemie.uni-giessen.de (S.M.); jonas.schmidt@chemie.uni-giessen.de (J.S.); holger.zorn@uni-giessen.de (H.Z.); 2Institute of Food Chemistry and Food Biotechnology, Justus Liebig University, 35392 Giessen, Germany

**Keywords:** black soldier fly, palm kernel meal, insect rearing, culturable microbiome, infection, core microbiome, entomopathogens

## Abstract

Larvae of the black soldier fly (BSF) *Hermetia illucens* are polyphagous feeders and show tremendous bioconversion capabilities of organic matter into high-quality insect biomass. However, the digestion of lignocellulose-rich palm oil side streams such as palm kernel meal (PKM) is a particular challenge, as these compounds are exceptionally stable and are mainly degraded by microbes. This study aimed to investigate the suitability of BSF larvae as bioconversion agents of PKM. Since the intestinal microbiota is considered to play a key role in dietary breakdown and in increasing digestibility, the bacterial and fungal communities of BSF larvae were characterized in a culture-dependent approach and screened for their putative entomopathogenicity. The lethality of six putative candidates was investigated using intracoelomal injection. In total, 93 isolates were obtained with a bacterial share of 74% that were assigned to the four phyla *Actinobacteria*, *Bacteroidetes*, *Firmicutes*, and *Proteobacteria*. Members of the genera *Klebsiella*, *Enterococcus,* and *Sphingobacterium* are part of the core microbiome, as they were frequently described in the gut of *Hermetia* larvae regardless of diet, nutritional composition, or rearing conditions. With 75%, a majority of the fungal isolates belonged to the phylum *Ascomycota*. We identified several taxa already published to be able to degrade lignocelluloses, including *Enterococcus*, *Cellulomonas*, *Pichia* yeasts, or filamentous *Fusarium* species. The injection assays revealed pronounced differences in pathogenicity against the larvae. While *Alcaligenes faecalis* caused no, *Diutina rugosa* weak (23.3%), *Microbacterium thalassium* moderate (53.3%), and *Pseudomonas aeruginosa* and *Klebsiella pneumoniae* high (≥80%) lethality, *Fusarium solani* injection resulted in 100% lethality.

## 1. Introduction

*Hermetia illucens* (*Diptera*: *Stratiomyidae*; BSF), commonly known as the black soldier fly, is native to the tropical and subtropical regions of America. Due to anthropogenic influences, it has spread almost worldwide [1]. In addition to a wide tolerance range for environmental factors, their short life cycle of 40–45 d makes the fly an interesting candidate for industrial applications [2]. Based on the polyphagous diet of the larvae, they have an extremely wide substrate spectrum. Various studies report their bioconversion capabilities of different organic matter and waste such as kitchen scraps, pineapple peelings, soybean residue [3], brewer’s spent grains [4], fish offal [5], as well as chicken and swine manure [6,7] into high-quality insect biomass. In particular, the high protein and fat content of the larvae are promising for novel uses in the feed sector, as an alternative to fish and soy meal formulations in aquaculture or livestock, respectively [8].

The potential to transform byproducts of agroindustry, livestock, or urban wastes into a sustainable protein source goes hand in hand with possible chemical and microbiological risks that need to be assessed with regard to feed and food safety [9]. Studies demonstrated that the composition of the gut microbial community, including bacteria and fungi, is strongly influenced by the feeding substrate [10,11] and numerous extrinsic parameters [2]. The microbiota harbored by insects are involved in complex processes such as the development of the gastrointestinal tract, modulating the immune system [12], the digestion of plant polymers [13], synthesizing essential amino acids and vitamins [14], pheromone and kairomone synthesis for inter- and intraspecific communication, as well as the defense against pathogen and parasite colonization [15]. In particular, the ability to grow on substrates with a high microbial load, such as feces, implies a highly adaptable immune response. BSF larvae were even able to reduce the concentration of prominent pathogens such as *Salmonella* spp. in human feces or the enterohemorrhagic *Escherichia coli* O157:H7 in chicken manure [16,17]. In contrast, *Salmonella enteritidis* and the food pathogen *Bacillus cereus* could accumulate in the gut of larvae grown on food waste or chicken manure, respectively [16,18]. The mechanistic role of the microbiome in the defense against unwanted bacteria and fungi is still unclear. However, a direct connection between the gut microbial community, the feeding substrate, and the expression of more than 50 putative antimicrobial peptides has already been demonstrated [19]. The inoculation of the substrate with companion bacteria also had positive effects on the growth and developmental parameters of black soldier fly larvae [20].

The intestinal microbiota of the black soldier fly probably contributes heavily to diet breakdown, e.g., through the degradation of plant polymers such as cellulose or lignin, and enables the larvae to utilize a variety of substrates [10]. Concurrently, such a microbial enzyme repertoire offers an exploitation potential for biotechnological applications. Based on this hypothesis, fiber-rich organic byproducts of the palm oil industry could be suitable as a potential feeding regime for the larvae. Palm oil production is one of the fastest-growing industries, mainly due to its comparatively high space efficiency and the steadily increasing demand for vegetable oils worldwide, resulting in the deforestation of large rainforest areas that are being replaced by oil palm (*Elaeis guineensis*) monocultures [21,22]. In 2018, 71.5 million tons of palm oil were produced on a cultivated area totaling 18.9 million ha, of which the main producers, Indonesia and Malaysia, have a share of 84% [23]. With a yield of 20–23% crude palm oil from the fruits, millions of tons of organic byproducts are generated annually, including empty fruit bunches (EFB), palm kernel meal (PKM), or the liquid palm oil mill effluent (POME). Due to its cost efficiency and high fiber content, PKM is mainly used in ruminant diets but is also used successfully as a supplement for monogastric livestock such as poultry and pigs. PKM was also established as feed for BSF larvae [24,25,26]. The objectives of the present work, therefore, were to isolate and determine the composition of the cultivable intestinal microbiota of BSF larvae and its function in the digestion of PKM. In terms of larval health and yield, putative entomopathogens of BSFs were characterized and identified for the first time.

## 2. Materials and Methods

### 2.1. Rearing of Hermetia illucens

BSF larvae were provided by Bio.S Biogas (Grimma, Germany). The larvae were fed *ad libitum* with PKM and maintained in 19.5 × 16.5 × 9.5 cm (l × w × h) polypropylene containers with a density of 150 mg eggs per container at 27 ± 1 °C and 60 ± 10% relative humidity (RH) in darkness [3,27,28,29]. The PKM was provided by PT Alternative Protein Indonesia (Tebet, Indonesia) and stored in a dry and dark place until it was fed. The developmental stage was determined by the weight (AT261 DeltaRange, Mettler, Giessen, Germany), length, and head capsule width (Keyence VHX-2000 digital microscope, Keyence, Osaka, Japan) as described elsewhere [30].

### 2.2. Cultivation of Bacterial and Fungal Isolates

In order to cultivate gut-associated microbiota, four agar media were used: LB medium (10 g L^−1^ tryptone, 5 g L^−1^ yeast extract, 5 g L^−1^ NaCl, 15 g L^−1^ agar) and TSA medium (17 g L^−1^ casein peptone, 2.5 g L^−1^ K_2_HPO_4_, 2.5 g L^−1^ glucose, 5 g L^−1^ NaCl, 3 g L^−1^ soy peptone, 15 g L^−1^ agar) for bacterial isolates, and YPD medium (20 g L^−1^ peptone, 10 g L^−1^ yeast extract, 20 g L^−1^ glucose, 20 g L^−1^ agar) and M_2_ medium (20 g L^−1^ malt extract, 3 g L^−1^ yeast extract, 15 g L^−1^ agar) for fungal isolates [11]. Fungal growth media were supplemented with 50 mg L^−1^ chloramphenicol.

Sixteen L5 larvae (150 ± 25 mg) were collected from different container positions with a spring steel tweezer and washed with distilled water. After anesthesia at −20 °C, their surface was sterilized twice with 70% ethanol (*v*/*v*). The guts were dissected under a stereomicroscope (S9i, Leica Microsystems, Wetzlar, Germany), weighed, and washed in 0.9% NaCl solution (*w*/*v*). Homogenization was carried out individually with sterile glass pestles in 300 µL LB or fungal enrichment GLY medium (20 g L^−1^ glycerol, 10 g L^−1^ yeast extract), respectively. Ten-fold serial dilutions were produced in 0.9% NaCl (*w*/*v*) solution, of which 100 µL were plated onto the four media mentioned above and incubated aerobically at 27 °C (bacteria) or 24 °C (fungi) in darkness. Colony-forming units (CFU) or conidia were calculated after 2 (bacteria), 4 (yeasts), and 7 d (filamentous ascomycetes) to obtain initial information about the microbiota harbored in a sample. Growing colonies and mycelia were selected based on their morphology.

To obtain pure cultures, isolates were transferred twice to fresh agar plates. For DNA extraction, liquid LB/TSA or YM broth (5 g L^−1^ peptone, 3 g L^−1^ malt extract, 3 g L^−1^ yeast extract, 10 g L^−1^ glucose, 50 mg L^−1^ chloramphenicol) for fungi were inoculated with pure cultures [11] and incubated shaking at 180 rpm and 27 °C for bacteria or 150 rpm and 24 °C for fungi. Bacterial and fungal isolates were stored in 25 or 10% glycerol at −80 °C, respectively.

### 2.3. DNA Extraction

In total, 1–2 mL of bacterial liquid cultures were centrifuged at 15,000 relative centrifugation force (rcf) for 5 min at room temperature (RT), whereas 15 mL of fungal liquid cultures were centrifuged at 1400 rcf for 10 min at 4 °C. Bacterial DNA was isolated from the resulting pellet using the PureLink^®^ Genomic DNA mini kit (Invitrogen, ThermoFisher Scientific, Waltham, MA, USA) using the protocol specifications for gram-positive cell lysates.

A total of 250 µL of yeast pellets or one micro spatula of filamentous mycelia were ground under liquid nitrogen, transferred into a 1.5 mL centrifugation tube, resuspended in 500 µL lysis buffer (400 mM Tris HCl, 60 mM ethylenediaminetetraacetic acid (EDTA), 150 mM NaCl, 1% sodium dodecyl sulfate) and vortexed. After an incubation of 10 min at RT, 150 µL 3 M potassium acetate solution was added, vortexed, and centrifuged at 17,500 rcf for 10 min at RT. The supernatant and the same volume of isopropanol was then transferred into a new tube and centrifuged under the same conditions to precipitate the DNA. The obtained DNA pellet was washed with 70% ethanol (*v*/*v*) and centrifuged for 1 min under the same conditions.

Purified chromosomal DNA was quantified by NanoDrop 2000 (ThermoFisher Scientific, Waltham, MA, USA) and stored at −20 °C until use.

### 2.4. 16S rRNA and ITS PCR

For species identification, 16S rRNA and 5.8S-ITS regions were amplified with the following primer pairs: 27F (5′-GGT TAC CTT GTT ACG ACT T-3′), 1492R (5′-AGA GTT TGA TCM TGG CTC AG-3′) [31] and ITS1 (5′-TCC GTA GGT GAA CCT GCG G-3′), ITS4 (3′-TCC TCC GCT TAT TGA TAT GC-3′) [32]. PCR was carried out in 25 µL reactions containing 0.5 pM of each primer, 0.2 mM dNTPs, 1 × Phusion GC buffer, Phusion polymerase (ThermoFisher Scientific, Waltham, MA, USA), and 100 ng template DNA. The cycling conditions for 16S rRNA amplification were set to: initial denaturation at 98 °C for 2 min, 35 cycles at 98 °C for 30 s, 58 °C for 30 s, 72 °C for 45 s, and final elongation at 72 °C for 5 min. 5.8S-ITS region was amplified under the following conditions: initial denaturation at 94 °C for 2 min, 35 cycles at 94 °C for 30 s, 51 °C for 30 s, 72 °C for 15 s, and a final elongation for 5 min at 72 °C.

Approximately 5 µL of the PCR products were then separated electrophoretically in a 1% agarose gel in 1 × TAE buffer (40 mM Tris base, 20 mM acetic acid, 1 mM EDTA) for 30 min at 125 V. Midori Green Advance (Nippon Genetics, Dueren, Germany) served as in-gel DNA stain. Each amplicon was sequenced by Sanger sequencing with the corresponding forward and reverse primers (Microsynth, Balgach, Switzerland). The resulting sequences were analyzed by the Nucleotide BLAST algorithm (http://www.ncbi.nlm.nih.gov/blast, accessed on 16 December 2021) for highly similar sequences in the nucleotide collection (nr/nt) database. A sequence identity of 97% was defined as the threshold. Subsequently, taxonomic classification was performed based on the NCBI Taxonomy Browser (https://www.ncbi.nlm.nih.gov/Taxonomy/Browser/wwwtax.cgi, accessed on 11 October 2021). In addition to morphological characteristics, isolates of the same species were compared genetically by pairwise alignment (Geneious 9.1.8, Biomatters, Auckland, New Zealand). If differences in the sequences were identified, these were retained as a different genotype.

### 2.5. Growth Curves, OD_600_/CFU Relationship, and Antibiotic Susceptibility Tests of Putative Entomopathogens

An extensive literature review was conducted to condense putative candidates from our bacterial and fungal isolates, which were previously reported to have entomopathogenic potential. In order to obtain synchronized bacterial or yeast cultures, single colonies of putative entomopathogens were inoculated in 20 mL liquid LB, TSA, or YPD, depending on the medium on which they were isolated (Appendix A). A representative isolate with the highest sequence identity was selected and used for further infection experiments. Isolates were deposited at the German Collection of Microorganisms and Cell Cultures GmbH (DSMZ, Braunschweig, Germany): *Microbacterium thalassium* (DSM 112768), *Diutina rugosa* (DSM 112794), *Fusarium solani* (DSM 112793), *Alcaligenes faecalis* (DSM 112765), *Klebsiella pneumoniae* (DSM 112766), and *Pseudomonas aeruginosa* (DSM 112767). The cultures were grown overnight at 27 °C and 180 rpm. In total, 500 µL of those cultures were transferred into 100 mL fresh medium and incubated under the same conditions. Growth curves were measured in 30 min intervals until an OD_600_ of 1.0 was reached (Ultrospec 10, Biochrom, Berlin, Germany). At least five samples of 1 mL (OD_600_ = 0.2–1.0) were taken, serially diluted in 0.9% NaCl (*w*/*v*) solution and 100 µL were plated onto the corresponding agar medium. After an incubation of 2 d at 27 °C, CFU·mL^−1^ was calculated. Experiments were carried out as biological triplicates.

Antibiotic susceptibility tests, including the determination of the minimal inhibitory concentration (MIC) for *P. aeruginosa* and *K. pneumoniae* were performed with the VITEK 2 system (BioMérieux, Marcy-l’Étoile, France); susceptibility analysis of *A. faecalis* and *M. thalassium* were performed by quantitative Etest (bestbion dx, Cologne, Germany). For resistance detection, 100 μL of log-phase liquid cultures (OD_600_ = 0.6) were plated on LB agar. The test strip was placed in the center of the plates. After an incubation of 2 d at 27 °C, zones of inhibition were examined. The results were interpreted in accordance with the guidelines of the European Committee on Antimicrobial Susceptibility Testing (EUCAST) and the Clinical and Laboratory Standards Institute (CLSI) [33,34].

### 2.6. Injection Assay

Three single colonies of putative entomopathogenic bacteria or yeasts were inoculated in liquid media and incubated as described above until they reached the log phase. OD_600_ was determined, and 1 mL of the liquid culture was washed twice in sterile 0.9% NaCl (*w*/*v*) solution to avoid interference from the medium. The density was then adjusted to 2 × 10^8^ CFU·mL^−1^ using the linear regression function. In order to achieve a comparable infection dose with the filamentous ascomycete *F. solani*, YPD agar plates were inoculated with a disc (0.6 cm) of an actively growing, one-week-old *F. solani* culture, and incubated at 24 °C for 12 days until sporulation occurred. For infection, *F. solani* conidia were released and harvested by rubbing the surface of plates covered with extraction solution (0.9% NaCl, 0.05% Tween-20) using a sterile Drigalski spatula [35]. Spores were separated completely from larger fragments of the hyphae by glass wool filtration, followed by two washing steps similar to the bacteria procedure and resuspended in 1 mL 0.9% NaCl solution. The number of conidia was determined by a hemocytometer and adjusted to 2 × 10^8^ conidia·mL^−1^ [36].

Ten L5 larvae of similar weight (150 ± 20 mg) from three breeding containers were selected at random. Each experimental group was surface sterilized twice with 70% ethanol (*v*/*v*) before being infected. A total of 5 µL containing 1 × 10^6^ CFU or conidia were ventrally injected with insulin syringes (Micro-Fine + U-100, Becton Dickinson, Franklin Lakes, NJ, USA) in the haemocoel of the third thoracic segment of the larvae [37]. The same number of non-infected and 0.9% NaCl-injected larvae served as negative controls. Post-infection, the larvae were maintained in 19.5 × 16.5 × 9.5 cm (l × w × h) polypropylene containers at 27 ± 1 °C and 60 ± 10% RH in darkness. They were fed with 20 g chicken feed (16% crude protein, 3.5% crude fat, 5% crude fiber, 12.5% crude ash; GoldDott Eierglück, DERBY Spezialfutter, Muenster, Germany) per container. The lethal time (LT_50_; time until 50% of the individuals were dead) and percent of mortality were monitored in 24 h intervals over 7 d (bacteria, yeast) or 10 d (*F. solani*) post-infection by exposing the larvae to mechanical stimuli. If they did not respond, they were considered dead [38]. Experiments were carried out as biological triplicates.

### 2.7. Data Processing

Data were processed with Excel 2016 (Microsoft, Redmond, WA, USA) and graphed with OriginPro 2020b (OriginLab Corporation, Northampton, MA, USA). The 16S rRNA and 5.8S-ITS sequences were used to create phylogenetic trees by multiple sequence alignment with the ClustalW software implemented in Geneious [39]. The trees were built using the neighbor-joining method and the JC69 Jukes and Cantor substitution model with 1000 bootstrap replications. β-diversity was calculated using the Jaccard index (I_J_). A simple linear regression model of OD_600_ and log_10_(CFU) was performed to describe the relationship between the photometrically measured density and the viable cell count. Lifetime data post-infection were analyzed by the Kaplan–Meier estimation to generate *S*(*t*) survival functions, which were compared pairwise by log-rank test and an error level of α = 0.05 for statistical significance.

## 3. Results

### 3.1. Taxonomic Composition of the Culturable Gut Microbiota in BSF Larvae Grown on PKM

The cultivable microbial gut community of BSF larvae reared on PKM was predominantly composed of bacteria. With ≥7 × 10^8^ CFU·gut^−1^, these showed a 100 to 100,000-fold higher abundance of cultivable cells than yeasts or filamentous ascomycetes, respectively (Table 1).

From the total of 16 dissected guts, 138 isolates were obtained (82 bacterial and 56 fungal) on the basis of morphological characteristics. After pairwise comparison, we identified 93 isolates with unique 16S rRNA (Appendix A) or 5.8S-ITS sequences (Appendix A), indicating that the sequences of 45 isolates (32.6%) are redundant. Consequently, these isolates were discarded. All sequences showed 98–100% identity with sequences available in the NCBI nucleotide collection database; isolates with a sequence identity <100% were integrated into the database. The detailed results of the classification and corresponding accession numbers are shown in Appendix A. Approximately 74% of the isolates were bacteria; the remaining species were fungi. In total, 51% of the bacteria can be assigned to Gram-positive, 38% to Gram-negative, 7% to Gram-variable, and 4% to taxa that cannot be determined by Gram staining (Figure 1). Overall, we were able to clearly identify 15 bacterial and 7 fungal taxa down to the species level, whereby 53 isolates of both domains were identified down to the genus level on the basis of similar sequence identities with at least two or more database entries.

The 69 bacterial isolates were assigned to the four phyla *Actinobacteria* (11 isolates), *Bacteroidetes* (nine isolates), *Firmicutes* (21 isolates), and *Proteobacteria* (25 isolates), which are categorized into 7 classes, 9 orders, 16 families, and 23 genera. We found no representatives of other phyla. Accordingly, a new genus was identified with every third isolate. Three prokaryotic isolates could only be assigned to the domain of the bacteria. The α diversity, according to Whittaker describing the diversity within a habitat, showed strong differences between the TSA (46 isolates) and the LB medium (27 isolates). In addition, the two media only had a Jaccard index of *I_J_* = 0.07, suggesting a low isolate similarity. More than 66% of the bacteria belonged to the *Proteobacteria* and *Firmicutes*, with a proportion of the isolates of 36% and 30%, respectively (Figure 2).

The phylogenetic relationship within the bacterial community was calculated using the obtained sequences and illustrated in a neighbor-joining tree (Figure 3). Among the most prominent phylum *Proteobacteria*, most isolates belonged to the families *Alcaligenaceae* (10 isolates) and *Enterobacteriaceae* (9 isolates) and were assigned to the genera *Klebsiella* (7 isolates), *Bordetella* (5 isolates), *Alcaligenes* (3 isolates *A. faecalis*), *Achromobacter* (2 isolates), and *Citrobacter* (2 isolates *C. amalonaticus*). Three isolates each belonged to the families *Pseudomonadaceae* (all assigned to the genus *Pseudomonas*) and *Brucellaceae*, which were assigned to the genera *Ochrobactrum*, *Brucella,* and *Bordetella*. The genera *Pseudomonas* (all isolates), *Achromobacter* (both isolates), and *Brucella* were only isolated from TSA agar, whereas *Ochrobactrum* was the only genus that was obtained exclusively from LB agar (Appendix A). *Bordetella* sp. (five isolates), *Klebsiella pneumoniae* (one to five isolates), and *Pseudomonas aeruginosa* (one isolate) were identified as prominent human pathogens.

Within the phylum *Firmicutes*, nine isolates belonged to the family *Enterococcaceae* (all genus *Enterococcus*), eight isolates to the family *Paenibacillaceae* (genus *Paenibacillus* and *Cohnella*), three isolates to the family *Bacillaceae* (all genus *Bacillus*), and one to the family *Lactobacillaceae* (genus *Lactobacillus*). Representatives of the genera *Bacillus*, *Lactobacillus* and *Cohnella* were cultivated exclusively on TSA agar, while all other *Firmicutes* isolates were obtained from cultures grown on TSA and LB agar plates.

The other phyla, with 16% (*Actinobacteria*) and 13% (*Bacteroidetes*), were represented significantly weaker in larval guts. Interestingly, all representatives of the *Actinobacteria* can be assigned to the order *Micrococcales*, whereby four isolates belonged to the family *Micrococcaceae* (genus *Kocuria* and *Micrococcus*), three isolates to the family *Microbacteriaceae* (genus *Microbacterium* and *Leucobacter*), two isolates to the family *Cellulomonadaceae* (both *Cellulomonas flavigena*), one isolate to the family *Dermacoccaceae* (*Dermacoccus nishinomiyaensis*), and one to the family *Promicromonosporaceae* (genus *Cellulosimicrobium*). Furthermore, the isolates of the genera *Dermacoccus*, *Leucobacter*, *Kocuria,* and *Cellulosimicrobium* were exclusively obtained from TSA agar plates.

Within the phylum *Bacteroidetes*, five isolates were assigned to the family *Sphingobacteriaceae* (all genus *Sphingobacterium*) and four to the family *Flavobacteriaceae* (genus *Empedobacter* and *Flavobacterium*) (Figure 3). All *Bacteroidetes* genera were obtained from both LB and TSA agar plates. Despite chloramphenicol supplementation (50 mg·L^−1^), *Sphingobacterium thalpophilum* (isolate 1) could be identified on M_2_ medium in addition to TSA.

The 24 fungal isolates were classified into 3 phyla, 4 classes and orders, 7 families, and the same number of genera (1 genus from every 3.4 isolates; Figure 2, Figure 4). We found no representatives of other phyla. The fungal growth media M_2_ (12 isolates) and YPD (10 isolates) reached the highest α diversity, followed by the bacterial growth media TSA (6 isolates) and LB (5 isolates). TSA and LB showed the highest similarity with a Jaccard index of *I_J_* = 0.43, while isolates were grown on M_2_ and YPD (*I_J_* = 0.24), M_2_ and TSA (*I_J_* = 0.18), TSA and YPD (*I_J_* = 0.11), as well as M_2_ and LB (*I_J_* = 0.10) had noticeably lower similarities. YPD and LB had no similarity (*I_J_* = 0).

With 75%, a majority of the fungi belonged to the phylum *Ascomycota* (18 isolates) distributed in the orders *Hypocreales* (three isolates) and the dominant *Saccharomycetales* (15 isolates), which contained 63% of the total fungal isolates. Within the phylum *Ascomycota*, seven isolates were assigned to the family *Saccharomycetaceae* (genus *Diutina* and *Kluyveromyces*), four isolates to the family *Pichiaceae* (two isolates *Pichia kudriavzevii* and two isolates *Pichia/Suhomyces*), three isolates to the family *Debaryomycetaceae* (all *Candida tropicalis*), three isolates to the family *Nectriaceae* (all genus *Fusarium*), and one to the family *Dipodascaceae* (*Sporopachydermia lactativora*). Representatives of the families *Saccharomycetaceae* and *Debaryomycetaceae* were isolated from a variety of different media, including YPD, M_2_, LB, and TSA agar. In contrast, *S. lactativora* was exclusively obtained from a culture grown on YPD agar (Appendix A).

The phylum *Basidiomycota* was only represented by five isolates, which were assigned to the family *Trichosporonaceae* (all *Trichosporon asahii*). *T. asahii* was isolated on YPD, M_2_, and TSA agar. Except for the *Nectriaceae* family, all members of the *Ascomycota* and *Basidiomycota* were yeasts (83% of total fungal isolates; Figure 1B).

The phylum *Mucoromycota* was exclusively represented by a single isolate, which was assigned to the family *Lichtheimiaceae* (*Lichtheimia ramosa*) and obtained from a culture grown on M_2_ agar (Figure 4, Appendix A).

### 3.2. Characterization of Putative Entomopathogenic Candidates from BSF Guts

An extensive literature review was conducted to condense putative candidates from our bacterial and fungal isolates, which were previously reported to have entomopathogenic potential. Four bacteria and two fungal species isolated from the BSF larval gut have already been described as putative insect pathogens in the literature: *Microbacterium thalassium* for *Ostrinia nubilalis* (*Lepidoptera*: *Crambidae*) [40], *Alcaligenes faecalis*, *Klebsiella pneumonia*, and *Pseudomonas aeruginosa* for *Galleria mellonella* (*Lepidoptera*: *Pyralidae*) [41,42,43], as well as *Diutina rugosa* for *Anastrepha ludens* (*Diptera*: *Tephritidae*) [36,44] and *Fusarium solani* for *Dendroctonus frontalis* (*Coleoptera*: *Curculionidae*) [45].

Isolates were deposited at the German Collection of Microorganisms and Cell Cultures GmbH (DSMZ, Braunschweig, Germany): *Microbacterium thalassium* (DSM 112768), *Diutina rugosa* (DSM 112794), *Fusarium solani* (DSM 112793), *Alcaligenes faecalis* (DSM 112765), *Klebsiella pneumoniae* (DSM 112766) and *Pseudomonas aeruginosa* (DSM 112767).

First, the four bacterial candidates were examined for possible antibiotic resistance by the VITEK 2 system and the quantitative Etest. All tested strains were susceptible to meropenem and trimethoprim/sulfamethoxazole. The antibiotic susceptibility test showed resistance to rifampicin and the *β*-lactam antibiotics ampicillin and piperacillin, but no evidence on extended-spectrum *β*-lactamases (ESBL) in *K. pneumoniae*, a member of the *Enterobacteriaceae* (Table 2). Cefuroxime had an intermediate effect on the growth of *K. pneumoniae*. *A. faecalis* had a high level of resistance to commonly used penicillins (ampicillin/sulbactam, piperacillin/tazobactam) and cephalosporins, including cefepime, cefotaxime, and ceftazidime. In addition, *A. faecalis* showed intermediate susceptibility against imipenem and rifampicin. *P. aeruginosa* was resistant to astreonam, piperacillin/tazobactam and rifampicin while being intermediate susceptible to cefepime, ceftazidime, ciprofloxacin, imipenem, and piperacillin. Antibiotics penetrating Gram-positive bacteria were chosen in accordance with CLSI guidelines for infrequently isolated and fastidious bacteria and an *M. paraoxydans* bacteraemia study [33,46]. *M. thalassium* showed resistance to cefepime, clindamycin, and rifampicin, while it was intermediate susceptible to cefotaxime, gentamicin, penicillin, and tetracycline. In total, three of the four isolates were resistant to rifampicin; the antimicrobial effect on *A. faecalis* was intermediate (Table 2).

In order to check their pathogenicity against BSF larvae, growth curves of the bacteria and yeast were first recorded and linear regressions of the CFU, and the corresponding OD_600_ were determined. *K. pneumoniae* reached an OD_600_ > 1 after 210 min, followed by *A. faecalis* (330 min). *M. thalassium* (450 min) took more than twice the time to reach OD_600_ = 1. Interestingly, *P. aeruginosa* and *D. rugosa* (both 390 min) showed a comparable growth phenotype (Figure 5).

The regression functions are given in Table 3. All liquid cultures had a comparable high regression coefficient of *R*^2^ > 0.98. The number of conidia of the filamentous ascomycete *F. solani* was determined by a hemocytometer.

### 3.3. In Vivo Evaluation of Putative Entomopathogens in BSF Larvae

To evaluate the pathogenic potential of fungal and bacterial species isolated from the gut of BSF larvae, injection assays were performed. All examined candidates, with the exception of *A. faecalis*, were able to cause an infection in the larvae within seven days post-injection. *A. faecalis* and *D. rugosa* demonstrated no change in survival probability compared to the NaCl-injected control group (*p* = 0.10).

The course of infection and the pathogenicity of *Klebsiella* and *Pseudomonas* species were comparably high (16.7% and 20.0% survival), albeit only *P. aeruginosa* caused the significantly greater killing of the larvae than the moderate *M. thalassium* (*p* = 0.02). Larvae injected with conidia of *F. solani* showed the lowest survival rate (*p* < 0.00001), with 100% lethality was already reached one day after inoculation (Figure 6A). In agreement, spores had germinated, and outgrowing mycelium indicated rapidly successful host colonization. An increase in the pigmentation of infected tissue due to melanization was observed in BSF larvae, similar to well-established insect infection models such as *G. mellonella* (Figure 6B).

## 4. Discussion

### 4.1. Analysis of the Cultivable Bacterial and Fungal Gut Microbiota in BSF Larvae Grown on PKM

The BSF has become the focus of science in recent years because its larvae are able to use almost all organic side streams as a feed substrate. The intestinal microbiome seems to play a key role in the dietary breakdown and the increase in digestibility, whereby PKM represents a particular challenge for microorganisms due to the high lignocellulose content (>20% dry matter) [10]. Above all, *Proteobacteria* such as *Rhizobiales*, *Burkholderiales*, *Enterobacteriales,* and members of the *Bacillales* (*Firmicutes*) are known for plant-associated nitrogen fixation [47]. However, studies suggest that *Rhizobiales* could also be endosymbioticly involved in nitrogen uptake in ants [48]. The putative role in the synthesis and provision of amino acids by *Rhizobiales* and *Burkholderiales* bacteria from the gut microbiome of BSF is already discussed elsewhere [49]. Members of the *Enterobacteriaceae*, especially *Enterobacter* or *Klebsiella*, are known for their ability to metabolize complex polysaccharides. The latter genus has already been identified frequently in the gastrointestinal tract (GIT) of BSF larvae reared on food waste, cooked rice, and calf forage and appears to be part of the core microbiota [10,18]. Isolates of *K. pneumoniae* from the GIT of *Bombyx mori* (*Lepidoptera*: *Bombycidae*) showed high *β*-endoglucanase and *α*-amylase activity. This points to the participation in the degradation of plant cell walls in the gut of BSF [50]. Typically, enterococci are also involved in the decomposition of plant polymers; they were detected as main actors in the digestion of lignocellulose in the longhorn beetle *Cyrtotrachelus buqueti* (*Coleoptera*: *Curculionidae*) [51]. Accordingly, the high relative abundance of nine *Enterococcus* isolates suggests a strong lignocellulolytic impact on PKM digestion in the larval GIT of BSF. Besides *Enterococcus*, the genus *Sphingobacterium* constitutes the core microbiome since both have been previously found in BSF larvae, regardless of the diet’s nutritional composition or rearing conditions [52,53,54,55]. Despite their low abundance, the two phenotypically different *Cellulomonas flavigena* isolates could, due to their cellulase and xylanase activities, make an important contribution to the degradation of cellulose contained in the PKM. In agreement with a study showing that a defined inoculation of companion *Bacillus subtilis* and *B. natto* strains in chicken manure had a positive effect on developmental time and growth performance of BSF, *Bacillus* isolates could consequently promote digestion of PKM [20]. The genera *Microbacterium*, *Micrococcus* and *Cellulomonas* belonging to the *Actinobacteria* were already isolated by subculturing from BSF eggs [56]. It is possible that these genera colonize adult flies and are transmitted vertically during oviposition. In this way, bacteria could be ingested from the egg surface during the early initial colonization of the hatched larvae and establish themselves in the GIT. Interestingly, the exposure of these three genera did not result in any olfactory responses from adult flies, whereas artificial oviposition sites inoculated with *Lactobacillus plantarum* were preferred by *Drosophila melanogaster* (*Diptera*: *Drosophilidae*) [56,57].

Until now, most of the publications on microbes have mainly focused on the bacterial composition in the intestine of BSF, while little is known about the dynamics and functional relationships of mycobiota. Comparison with the literature shows that, apart from *Mucoromycetes*, members of all classes have already been detected in the larval GIT [11]. With *Lichtheimia ramosa,* a member of the *Mucoromycetes* could be identified for the first time. Overall, yeasts represent the majority of fungal isolates with a relative abundance of more than 80%, which indicates an intensive insect-yeast association. Such beneficial interactions are widespread in insects and were frequently found in *Hymenoptera*, *Coleoptera,* and *Diptera* (especially fruitflies), which BSFs also belong to (reviewed in [58]). Several isolated yeasts, including *Candida*, *Pichia,* and *Trichosporon*, have already been detected in a culture-independent study in the gut of BSF larvae, which were grown on chicken feed or vegetable waste, respectively [11]. In particular, members of the *Saccharomycetales* such as *Candida* sp. and *Pichia* sp. are known to express antimicrobial peptides (AMPs), which are often directed against closely related yeasts and, thus, possibly protect the larvae against pathogen colonization [59]. Species of both genera (*C. tropicalis*, *P. kudriavzevii*) have also been described in palm wine from the oil palm therefore it is a reasonable assumption that they originate from the plant [60]. Further, *P. kudriavzevii* is described as a candidate for bioconversion of hemicellulosic materials into ethanol [61]. It is likely that *D. rugosa* (syn. *Candida rugosa*) is involved in the lipid degradation of palm (kernel) oil from PKM that has not been completely extracted during the milling process, as most frequently commercially used yeast lipases come from *Diutina* [62]. In agreement with PKM-fed larvae, *D. rugosa* was identified in substrate and frass samples of chicken feed and cottonseed press cake, suggesting a beneficial role in a functional or pathogen protective manner [63]. In addition to bacteria and yeasts, filamentous fungi of the phyla *Ascomycota* and *Basidiomycota* are known to produce various lignocellulolytic enzymes. These enzymes are secreted into their environment and, thus, making a significant contribution to the natural decay of plant biomass. Most ascomycetes are able to hydrolyze (hemi-) cellulose but are deficient in lignin degradation. The plant pathogen *Fusarium solani* f. sp. *glycines* (*Nectriaceae*), on the other hand, secretes laccases and lignin peroxidases, which enable lignin degradation and imply a similar function of *Fusarium* isolates in the GIT of BSF [64]. Slow degradation rates of lignin were also indicated for *F. solani* strains and other *Fusarium* sp. varieties [65]. This ability could also be related to the suspicion that *F. solani* is the causative agent of fatal yellowing disease in palms. Members of this genus could therefore be substrate endogenous and, consequently, have established themselves in the intestine of the larvae [66]. For *Monochamus marmorator* (*Coleoptera*: *Cerambycidae*), it could be shown that the ascomycete *Trichoderma harzianum* enables the beetle to degrade cellulose. Larvae of *M. marmorator* were only able to metabolize cellulose if *T. harzianum* was present in the feed. The fungus was not an integral part of the mycobiome but was only ingested through feed consumption [67]. Similar functions of mycobiota are also conceivable in larvae of BSF, which also ingest various fungi depending on the substrate [11].

Many studies reveal pronounced differences in the composition of the microbiome of BSF larvae, significantly affected by the diet. At the same time, Klammsteiner et al. (2020) and a comparison of our data with previously published community structures show that ingested substrates strongly influence the gut microbiome, but a non-impacted core community seems to be omnipresent [10,11,52,53,54,55].

### 4.2. Investigation of Putative Entomopathogenic Isolates

Since the BSF has only played an increasing role commercially for a few years, research on entomopathogens and their infection mechanisms have received little attention. In addition, many studies postulate an extraordinary colonization resistance against pathogens, which is mainly based on speculation and has not been adequately investigated. These hypotheses are mostly based on larval ability to live in environments such as feces, compost, or even carrion, representing particularly microbially contaminated regimes. Although there has not yet been a widespread outbreak in commercial breeding, the economic risk of diseases caused by entomopathogens has increased due to further upscaling of production [16,17,20,68,69]. For this reason, all bacterial and fungal isolates from the gut of PKM-fed larvae were examined for the mention of entomopathogenicity as part of an extensive literature search, whereby six putative candidates were identified [36,40,41,42,43,44,45]. The intracoelomal injection of putative entomopathogenic bacteria and fungi demonstrated pronounced differences in the survival probability. *A. faecalis* was not able to cause an infection in the larvae, suggesting an inefficient infection strategy against dipteran species. However, there is limited knowledge on the host spectrum of *A. faecalis*, as it was described only once in a study with *G. mellonella* [43]. The AflP-1A/1B binary toxin of *Alcaligenes* sp. was shown to be functionally homologous to the insecticidal Cry34Ab1/Cry35Ab1 proteins from *Bacillus thuringiensis*, inducing the perforation of the intestinal epithelium of coleopteran larvae such as *Diabrotica virgifera* (*Coleoptera*: *Chrysomelidae*) [70]. Even though *D. rugosa* caused an infection in a few BSF larvae, survival did not differ significantly from the NaCl-injected control. In accordance with our data, Salas et al. (2018) also categorized *D. rugosa* as a weakly pathogenic species in the Mexican fruit fly, *Anastrepha ludens*, due to a high yield of larvae and pupae grown in a yeast-inoculated diet [44]. *M. thalassium* caused moderate larval killing, which suggests the presence of virulence factors that allow the bacterium to bypass the larval immune response, but it does not appear to be a highly adapted system for *Diptera*. To date, no information on the infection mechanism is available in the literature, and merely one study demonstrated a moderate lethality in lepidopteran *Ostrinia nubilalis* larvae [40].

In contrast, *P. aeruginosa* and *K. pneumoniae* are well-studied in vertebrates, which is mainly due to the fact that they are prevalent nosocomial germs. Both species could, however, also be linked to the infection of *Diptera* such as *D. melanogaster* or *Aedes aegypti* (*Diptera*: *Culicidae*) and showed a high lethality of about 80% in BSF larvae [71,72]. In accordance, *Pseudomonas* was found frequently during early developmental stages on chicken feed, perceptible from the remarkable green substrate discoloration and the typical odor of linden blossoms (data not shown). Substrate colonization by *Pseudomonas* species may therefore contribute to lower larval survival rates at an early stage. In addition to fimbriae or pili, which enable adhesion to surfaces, *Klebsiella* and *Pseudomonas* are characterized by binding antimicrobial substances of the host immune system and protecting themselves lipopolysaccharide *O*-antigen-mediated from the opsonization by complement factor C3b. Moreover, exposed cell structures of *K. pneumoniae* are masked by a polysaccharide capsule, which protects it from phagocytosis [73,74]. *Pseudomonas* also has a broad repertoire of factors that allow it to damage the host. These lead to pore induction and perforation of cell membranes, as well as enzymatic modifications catalyzing depolymerization of actin filaments of the cytoskeleton or the ADP-ribosylation of elongation factor 2 by exotoxin A [75,76]. It is, therefore, a reasonable assumption that the high lethality resulted from the various virulence factors of both *Gammaproteobacteria*, which enable them to be highly adapted during different phases of infection (adhesion, invasion, establishment, reproduction, damage). Nevertheless, a recent study, which focused on the antibacterial effect of the HP/F9 peptide from the hemolymph of BSF larvae, revealed an effective *K. pneumoniae* growth inhibition in vitro and an in vivo reduction of inflammation in infected mice lungs [77]. BSF larval extracts also showed antibacterial activity against *P. aeruginosa* in substituted liquid cultures [68]. These findings indicate differences in pathogenicity and sensitivity to the immune response in a strain-dependent manner.

The inoculation of *F. solani* spores led to a tremendously high killing rate within a very short incubation time. Pathogenicity against BSF larvae is described here for the first time, although several members of the *Nematocera* such as *Culex pipiens* (*Diptera*: *Culicidae*), *Aedes cantans*, *A. detritus* (*Diptera*: *Culicidae*), and *Anopheles stephensi* (*Diptera*: *Culicidae*), as well as the brachyceran *Tetanops myopaeformis* (*Diptera*: *Ulidiidae*) were already killed by *Fusarium* sp. [78,79]. *Fusarium* species synthesize a broad portfolio of mycotoxins, including beauvericin, enniatins A, A1, B and B1, moniliformin, and fusaproliferin. Cyclodepsipeptides such as beauvericin and enniatin are able to integrate into the cell membrane. Due to their ring-shaped structure, they form cation-selective channels. In addition, beauvericin penetrates the cell nucleus and interacts with the host DNA, whereby beauvericin-DNA adducts can form [79,80]. Moniliformin, on the other hand, modifies various enzymatic processes in the host, which results in cell damage. The inhibition of glutathione-peroxidase and -reductase leads to an increase in oxidative stress; at the same time, transketolase and aldose reductase are also inhibited, which disrupts carbohydrate metabolism [80]. This remarkable variety of damaging secondary metabolites probably contribute significantly to the high lethality of infected larvae. Which mycotoxins are fundamentally essential for pathogenicity and how fungicidal substances affect survival rates of this highly virulent ascomycete should be clarified in further experiments.

The treatment of pathogens is of particular interest and might be difficult due to associated antibiotic resistance. Antibiotic resistance profiles of putative bacterial entomopathogens were determined by susceptibility tests, revealing several intermediate and resistant acting isolates. There is the possibility, similar to conventional livestock farming, to administer prophylactic antibiotics in insect breeding in order to prevent the spread of pathogens. However, this would also influence the beneficial microbiota of the BSF larvae, which may have an impact on growth performance and developmental rates. For example, the long-term medication of *Parasemia plantaginis* (*Lepidoptera*: *Erebidae*) larvae with fumagillin affected life-history traits, as well as their reproduction, negatively [81]. In addition, the widespread use of antibiotics has to be refused as it has been proven that it significantly contributes to the development of novel resistance mechanisms and promotes their spread. A more rational approach would be a targeted medication with specific antimicrobial substances as soon as a dominant pathogen has been detected in a breed, e.g., greening pseudomonads. The most efficient and gentle way would be the use of probiotics, which compete with pathogens for resources and habitat. However, further research is required to identify such probiotic microorganisms for BSF. Members of the genus *Bacillus* could be suitable probiotics since strains of *B. subtilis* and *B. natto* have already been positively associated with the development of BSF larvae [20]. It is conceivable that some yeasts are able to eliminate entomopathogenic closely related species by secreting AMPs [59]. The inoculation of core microbiota could also contribute to restoring the healthy balance of the microbiome by displacing pathogens. Furthermore, there is a possibility that antibiotic-resistant bacteria or their resistance determinants will be spread along the feed chain. Since BSF larvae are a promising alternative to fish meal and soy protein in livestock feeding, corresponding bacteria could colonize food and animals; thus reaching humans directly by interaction with animals and their excretions such as feces, urine, and saliva (especially farmers) or indirectly through consumption of contaminated food products [82,83].

### 4.3. Conclusions

In conclusion, our study shows that BSF larvae are suitable as a bioconversion agent for PKM, whereby the gut microbiota seems to play an important role. We identified several taxa that are able to degrade lignocelluloses, including *Enterococcus*, *Cellulomonas*, *Pichia* yeasts, or filamentous *Fusarium* species, which could increase the digestibility for the larvae. Moreover, the isolates generated could have the potential for industrial applications. As postulated by several studies, BSF larvae seem to have an adaptive immune system that enables them to survive even on substrates with high microbial contamination. On the other hand, our results show for the first time that both fungal and bacterial isolates from the gut of the larvae occasionally have a strong entomopathogenic potential when injected. Further research on the identification of dipteran entomopathogens and their infection mechanisms should be carried out. In the future, this could combat the spread of diseases caused by entomopathogens and prevent large monetary losses through the timely use of appropriate diagnostic protocols [69].

## Figures and Tables

**Figure 1 microorganisms-10-00319-f001:**
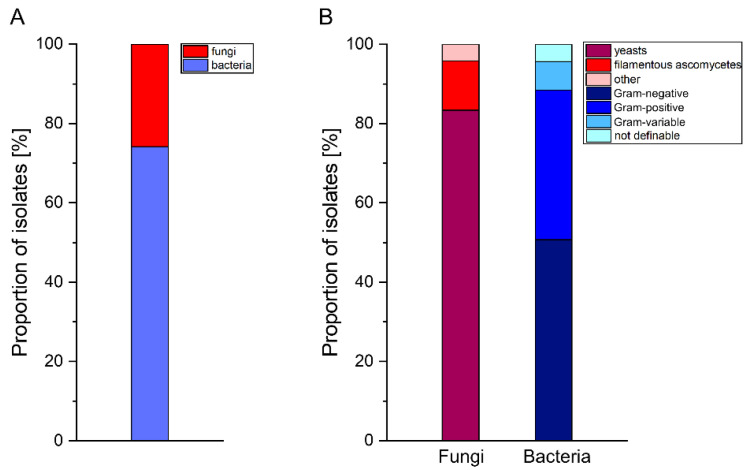
Superordinate categorization of the isolates. (**A**) Ratio of fungal and bacterial isolates. (**B**) left: Distribution of filamentous and yeast-like fungi. ‘Other’ describes a single member of the *Mucoromycota*. right: Bacterial community classification based on Gram staining characteristics.

**Figure 2 microorganisms-10-00319-f002:**
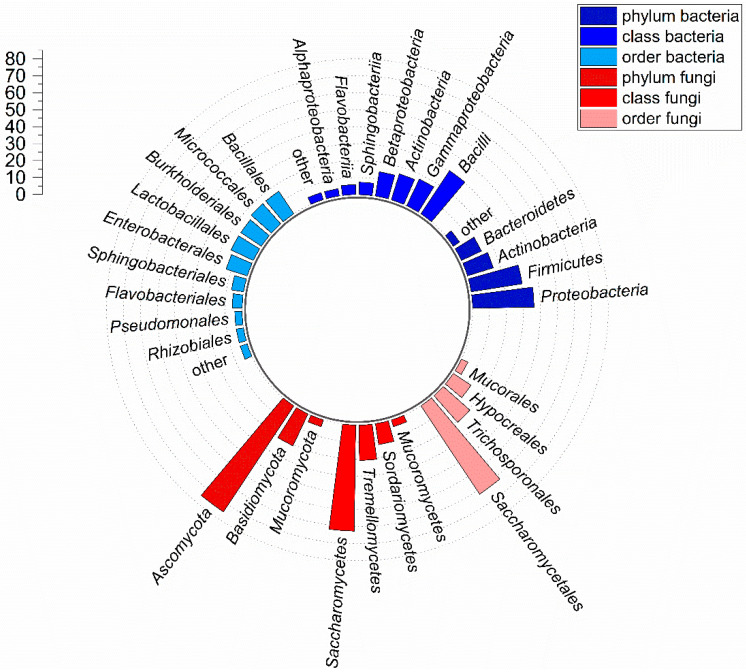
Radial bar plot of the cultivable gut microbiota composition. The proportion of bacterial and fungal phyla, classes and orders were grouped and compared. Dashed lines indicate 10% intervals. Isolates that could not be assigned to a specific taxon were summarized as ‘others’.

**Figure 3 microorganisms-10-00319-f003:**
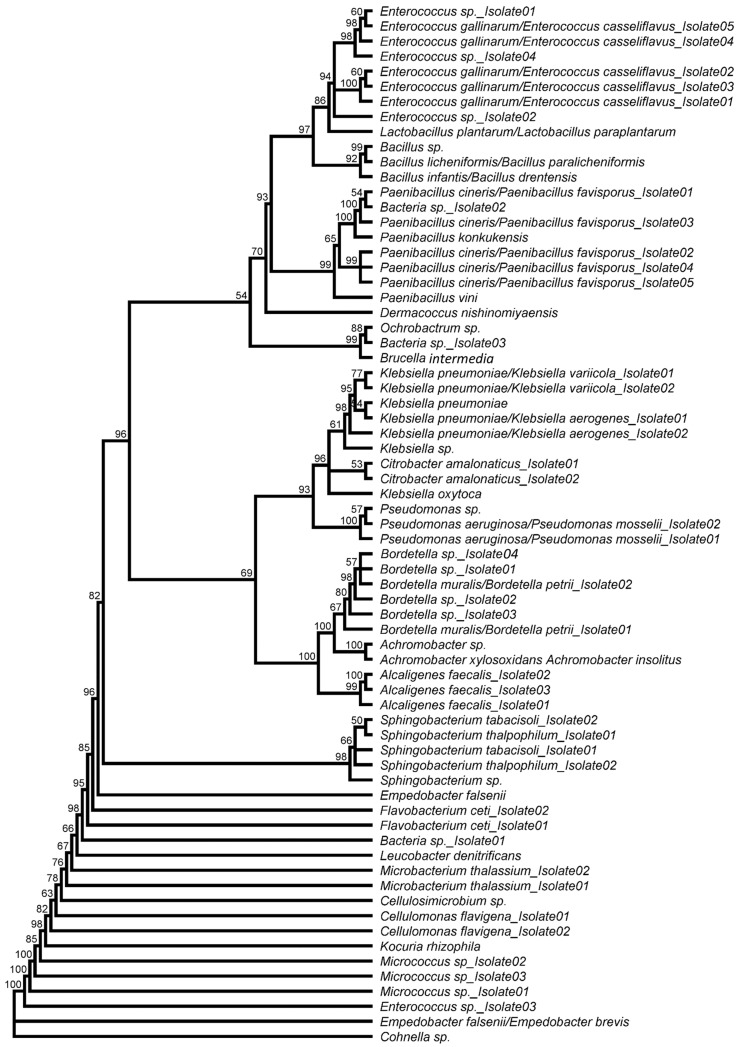
Phylogenetic analysis of all bacterial BSF gut isolates based on 16S rRNA sequences. Unrooted neighbor-joining tree using the JC69 Jukes and Cantor substitution model. The numbers indicate support for clade branching (%) of 1000 bootstrap replications.

**Figure 4 microorganisms-10-00319-f004:**
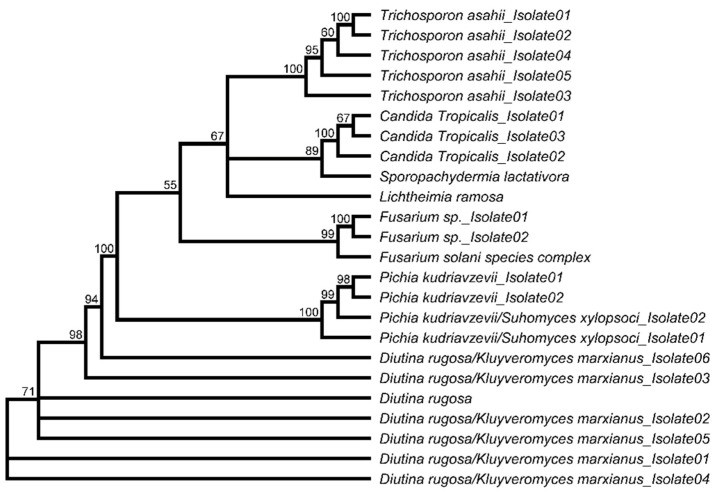
Phylogenetic analysis of all fungal BSF gut isolates based on 5.8S-ITS sequences. Unrooted neighbor-joining tree using the JC69 Jukes and Cantor substitution model. The numbers indicate support for clade branching (%) of 1000 bootstrap replications.

**Figure 5 microorganisms-10-00319-f005:**
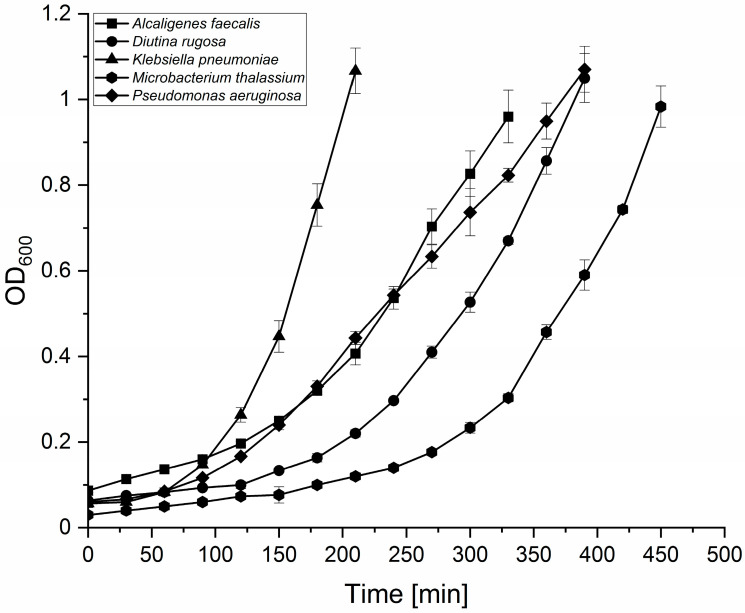
Growth curves of putative entomopathogenic bacterial (*A. faecalis*, *K. pneumoniae*, *M. thalassium*, *P. aeruginosa*) and yeast (*D. rugosa*) candidates. OD_600_ was measured in 30 min intervals; the average optical density OD_600_ (±SD) of three independent liquid cultures is shown.

**Figure 6 microorganisms-10-00319-f006:**
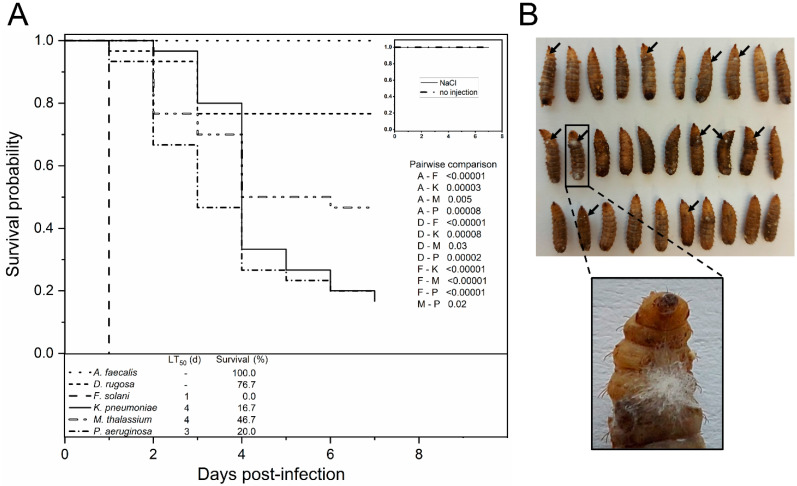
Inoculation of BSF larvae with putative entomopathogens leads to lethality in a species-dependent manner. (**A**) Kaplan-Meier survival functions of 30 larvae injected with 10^6^ CFU, conidia, or an equal volume of physiological NaCl (control). Not-injected larvae served as an additional control; larval survival was monitored daily. Lethal time is expressed as LT_50_ values (50% deaths). *p*-Values represent infections compared pairwise by log-rank test, with first letters indicating the genera. (**B**) Images of dead *F. solani* infected larvae one-day post-inoculation. Melanization of infected tissue and mycelium growing out of the injection site (black arrows) can be observed.

**Table 1 microorganisms-10-00319-t001:** Calculation of the bacterial and yeast CFU as well as the ascomycete conidia. Values represent the mean of 16 larval guts (±SD).

	Bacteria		Yeasts		Filamentous Ascomycetes	
	LB	TSA	YPD	M_2_	YPD	M_2_
CFU/conidia gut^−1^	1.06 × 10^9^	7.04 × 10^8^	1.10 × 10^6^	4.67 × 10^6^	1.40 × 10^4^	1.31 × 10^4^
(±SD)	(±9.81 × 10^8^)	(±4.63 × 10^8^)	(±3.22 × 10^5^)	(±2.83 × 10^6^)	(±7.87 × 10^3^)	(±9.44 × 10^3^)
CFU/conidia (mg gut)^−1^	1.35 × 10^7^	8.52 × 10^6^	1.47 × 10^4^	6.56 × 10^4^	1.90 × 10^2^	1.63 × 10^2^
(±SD)	(±1.39 × 10^7^)	(±5.48 × 10^6^)	(±3.90 × 10^3^)	(±4.34 × 10^4^)	(±1.11 × 10^2^)	(±1.03 × 10^2^)

**Table 2 microorganisms-10-00319-t002:** Antibiotic susceptibility test of putative bacterial entomopathogens. MIC (µg·mL^−1^) interpretative criteria were taken from the susceptibility breakpoint specifications of CLSI and EUCAST [33,34]. Interpretation abbreviations are defined as follows: susceptible (S), intermediate (I), resistant (R), insufficient evidence (IE), no suitable target (-), not determined (n).

Drug	Gram-Negative						Gram-Positive	
	*Alcaligenes faecalis*		*Klebsiella pneumoniae*		*Pseudomonas aeruginosa*		*Microbacterium thalassium*	
	MIC	Interpretation	MIC	Interpretation	MIC	Interpretation	MIC	Interpretation
Amikacin	n	n	n	n	≤2	S	(-)	(-)
Ampicillin	>256	R	≥32	R	(-)	(-)	>0.75	S
Ampicillin/ Subactam	≥64	R	≤2	S	(-)	(-)	n	n
Aztreonam	n	n	n	n	32	R	(-)	(-)
Cefepime	>48	R	n	n	4	I	>6	R
Cefotaxime	>256	R	≤1	S	(-)	(-)	≥2	I
Cefpodoxime	n	IE	≤0.25	S	(-)	(-)	n	IE
Ceftazidime	>256	R	≤1	S	4	I	n	IE
Cefuroxime	n	n	≤1	I	(-)	(-)	n	IE
Cefuroxime axetil	n	n	≤1	S	(-)	(-)	n	IE
Ciprofloxacin	≥0.19	S	≤0.25	S	0.5	I	≥0.5	S
Clindamycin	(-)	(-)	(-)	(-)	(-)	(-)	16	R
Colistin	n	n	n	n	≤0.5	S	(-)	(-)
Ertapenem	≥0.032	S	≤0.5	S	(-)	(-)	n	IE
Erythromycin	n	IE	n	n	N	n	≤0.5	S
Fosfomycin	n	IE	n	n	(-)	(-)	(-)	(-)
Gentamicin	<1.5	S	≤1	S	≤1	IE	≤12	I
Imipenem	≥2	I	≤0.25	S	2	I	≤0.5	S
Meropenem	≥0.38	S	≤0.25	S	≤0.25	S	≥0.125	S
Moxifloxacin	n	IE	≤0.25	S	(-)	(-)	n	IE
Penicillin	(-)	(-)	(-)	(-)	(-)	(-)	≥0.5	I
Piperacillin	n	n	8	R	16	I	n	IE
Piperacillin/ Tazobactam	>256	R	≤4	S	32	R	n	IE
Rifampicin	≥1.5	I	>32	R	8	R	≥12	R
Tetracyclin	n	IE	n	n	(-)	(-)	≤12	I
Tigecycline	≥0.5	S	n	n	(-)	(-)	n	IE
Tobramycin	n	n	n	n	≤1	S	(-)	(-)
Trimethoprim/ Sulfamethoxazole	≥1.5	S	≤20	S	(-)	(-)	>0.5	S
Vancomycin	(-)	(-)	(-)	(-)	(-)	(-)	>2	S/IE

**Table 3 microorganisms-10-00319-t003:** Linear regression model of log_10_(CFU·mL^−1^) and OD_600_ from liquid cultures of putative entomopathogenic bacterial (*A. faecalis*, *K. pneumoniae*, *M. thalassium*, *P. aeruginosa*) and yeast (*D. rugosa*) candidates.

Liquid Culture	n	Function	*R*²
*Alcaligenes faecalis*	5	y = 2.18x + 6.84	0.9954
*Diutina rugosa*	6	y = 1.57x + 6.80	0.9901
*Klebsiella pneumoniae*	5	y = 3.30x + 5.87	0.9985
*Microbacterium thalassium*	5	y = 2.79x + 6.25	0.9827
*Pseudomonas aeruginosa*	7	y = 1.76x + 6.87	0.9964

## Data Availability

All relevant data is contained within the article.

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
