# Peer review of "Isolation of Bacterial and Fungal Microbiota Associated with Hermetia illucens Larvae Reveals Novel Insights into Entomopathogenicity"

_microorganisms, 2022, doi:10.3390/microorganisms10020319_

Round 1
Reviewer 1 Report
The manuscript is interesting. It presents a very interesting topic about insect feed and their use as a feed product.
Title fits the content, relevant but a bit too long. It does not arouse the interest of readers.
List of authors and affiliations
This section is fine.
Abstract
In my opinion, this is the weakest part of the job. I have no information what was the purpose of the study? Why is the composition of the insect microflora so important for the digestion of plant material. The summary should be rewritten.
Keywords
They are OK. Be sure not to repeat words from the title.
Introduction
It is a valuable, well-structured introduction that clearly follows the path of thought. It provides a solid background and allows you to look at the study from a broader perspective. The research concept is clear, with well-defined hypotheses.
Materials and methods
Good description of the research area, sampling conditions and molecular work. Unfortunately, information on sequencing is missing. For the identification of fungi, the ITS region is not sufficient to define a taxon. Therefore, it is recommended to identify min. 3 regions and a common phylogenetic analysis. What tactic was used as an outgroup when creating trees?
Why were no statistical analyzes performed? Biodiversity indicators have not been calculated?
.
Results
This section is easy to follow and there are some interpretations as well.
Discussion
The results are discussed in a fairly good way, showing previous results from other studies. I like that this section is broken down into subsections by subject.
Minor errors, e.g. the authors state that the aspect concerns many publications, and then cite one ...
Line 416 The latter genus has already been frequently identified in the gastrointestinal tract (GIT) of BSF larvae grown on food waste, cooked rice and calf feed, and appears to be part of the underlying microflora [10].
Line 441 Until now, most of the publications on microbes have mainly focused on the bacterial composition in the intestine of BSF, while little is known about the dynamics and functional relationships of mycobiotics. Comparison with the literature shows that, apart from Mucoromycetes, samples of all classes of larvae have already been detected in GIT [11]
We do not start the sentence with an abbreviation: line 418 K. pneumoniae isolates from GIT Bombyx mori (Lepidoptera: Bombycidae) showed high -endoglucanase and -amylase activity. This points to the participation in the degradation of plant cell walls in the intestines of BSF [50]. T.
.
Conclusions
There is no clear summary
Bibliography
The number of references used is quite large. The manuscript is also based on contemporary literature published by reputable journals.
Supplementary materials
This section is fine.
Authors' contribution
This section is fine.
Acknowledgments - Financing - Conflict of Interest
These sections are fine.
Author Response
Reviewer #1
The manuscript is interesting. It presents a very interesting topic about insect feed and their use as a feed product. Title fits the content, relevant but a bit too long. It does not arouse the interest of readers.
-> We highly appreciated your comments and the work! The title has been shortened.
Abstract
In my opinion, this is the weakest part of the job. I have no information what was the purpose of the study? Why is the composition of the insect microflora so important for the digestion of plant material. The summary should be rewritten.
-> We thank Reviewer #1 for the helpful suggestions. The abstract has been rewritten to improve the understanding of our work. We added a clear purpose.
Keywords
They are OK. Be sure not to repeat words from the title
->improved accordingly
Materials and methods
Good description of the research area, sampling conditions and molecular work. Unfortunately, information on sequencing is missing. For the identification of fungi, the ITS region is not sufficient to define a taxon. Therefore, it is recommended to identify min. 3 regions and a common phylogenetic analysis. What tactic was used as an outgroup when creating trees?
-> Sequencing information was added. We are aware of the fact that the ITS region has drawbacks in several clades to clearly define the species level and we agree that for a taxonomic paper this would probably be not enough. However, our manuscript does not seek to function as a taxonomic paper, which can be clearly seen by the fact that our phylogenetic analysis only contains own isolates and sequences and no outgroup or reference sequences. In our opinion for the present study, the identification via ITS region suffice due to the fact that: i) the overall fungal analysis (Fig. 2) only depicts orders and ii) fungi whose ITS region shows high identity to more than one species are named with both names or as sp. (Table S2).
->When creating the trees, we used rooted neighbor joining methodology including the JC69 Jukes and Cantor substitution model where no outgroup was inserted.
Why were no statistical analyzes performed? Biodiversity indicators have not been calculated?
-> As we describe a gut community from a group of larvae grown on the same substrate, there is no statistical analysis necessary. Biodiversity indicators (like alpha/beta diversity) are helpful instruments to describe microbial communities. However, in our work we homogenized individual guts and plated them on four different media. Based on morphological characteristics, pure cultures were produced and sequenced what resulted in one community (not 16 communities). Cultures that matched optically and in DNA sequence were discarded. Therefore, diversity indicators can only be calculated between the different growth media, which were added in the manuscript.
Results
This section is easy to follow and there are some interpretations as well.
Discussion
The results are discussed in a fairly good way, showing previous results from other studies. I like that this section is broken down into subsections by subject.
Minor errors, e.g. the authors state that the aspect concerns many publications, and then cite one ...
Line 416 The latter genus has already been frequently identified in the gastrointestinal tract (GIT) of BSF larvae grown on food waste, cooked rice and calf feed, and appears to be part of the underlying microflora [10].
-> changed accordingly
Until now, most of the publications on microbes have mainly focused on the bacterial composition in the intestine of BSF, while little is known about the dynamics and functional relationships of mycobiotics. Comparison with the literature shows that, apart from Mucoromycetes, samples of all classes of larvae have already been detected in GIT [11]. We do not start the sentence with an abbreviation: line 418 K. pneumoniae isolates from GIT Bombyx mori (Lepidoptera: Bombycidae) showed high -endoglucanase and -amylase activity. This points to the participation in the degradation of plant cell walls in the intestines of BSF [50].
-> changed accordingly
Conclusions
There is no clear summary
->The conclusion was improved accordingly.
Reviewer 2 Report
Discussion is too lengthy, should be cut short, repetition of the introduction could be avoided.
Author Response
Discussion is too lengthy, should be cut short, repetition of the introduction could be avoided.
-> We thank the reviewer for this comment. We are aware that the conclusion is long, however not lengthy. As we subdivided the discussion section, the reader can focus on for him/her interesting parts. In our opinion, only the opening paragraph/sentence partly contains information from the introduction section. However, this introduction is needed to pick up and drive the reader towards the discussion. In addition, reviewer #1 valued the discussion section and its division positively.
Reviewer 3 Report
In the conclusions the importance of this research in the microbiota and future uses should be better explained
Author Response
In the conclusions the importance of this research in the microbiota and future uses should be better explained
-> We thank the reviewer for this comment! The importance and potential future use of the isolated microbiota have been added to the conclusion section.
Reviewer 4 Report
of the manuscript “Isolation of bacterial and fungal microbiota associated with Hermetia illucens larvae grown on palm kernel meal reveals novel insights into entomopathogenicity”
by authors Patrick Klüber, Jonas Schmidt, Stephanie Müller, Holger Zorn and Martin Rühl
The modern development of the agro-industrial complex leads to the accumulation of byproducts, which can be converted into a source of protein (feed and food). The palm oil production is one of the fastest growing industries. With a yield of 20-23% crude palm oil from the fruits, millions of tons of organic byproducts are generated annually, including palm kernel meal (PKM). The use of living organisms for the processing of these wastes is promising.
Fiber-rich organic byproducts of the palm oil industry could be suitable as a potential feeding regime for insect larvae. Especially high protein and fat content in larvae, in turn, makes it promising to use them as feed in aquaculture or animal husbandry. In this regard, the subject of the study is very relevant.
Larvae of the black soldier fly (BSF) Hermetia illucens are polyphagous and show tremendous bioconversion capabilities of organic matter into high quality insect biomass. BSF larvae are suitable for the use of vegetable by-products of palm oil production, such as lignocellulose-rich palm kernel flour. It is known that the intestinal microbiota of insects plays a key role in splitting food and increasing its digestibility. The authors were based on the hypothesis that the intestinal microbiota of the black soldier fly makes a special contribution to the decomposition of plant polymers such as cellulose or lignin. This study aimed to were to isolate and determine the composition of the cultivable intestinal microbiota of BSF larvae and its function in the digestion of PKM.
Methods of statistical analysis of the data obtained were applied. Data were processed with Excel 2016 (Microsoft) and graphed with OriginPro. The 16S rRNA and 5.8S-ITS sequences were used to create phylogenetic trees by multiple sequence alignment with the ClustalW software implemented in Geneious.
The taxonomic composition of the microbiome in the intestine of the black soldier fly was determined. The cultivable microbial gut community of BSF larvae reared on PKM was predominantly composed of bacteria. Authors were able to clearly identify 15 bacterial and seven fungal taxa down to the species level, and 53 isolates of both domains were identified down to the genus level. The 69 bacterial isolates were assigned to the four phyla Actinobacteria (11 isolates), Bacteroidetes (nine isolates), Firmicutes (21 isolates), and Proteobacteria (25 isolates). More than 66% of the bacteria belonged to the Proteobacteria and Firmicutes. The 24 fungal isolates were classified into three phyla, four classes and orders, seven families and the same number of genera. With 75%, a majority of the fungi belonged to the phylum Ascomycota. Several taxa capable of destroying lignocelluloses have been identified.
The bacterial and fungal communities of BSF larvae were screened for their putative entomopathogenicity. The lethality of six putative candidates was investigated using intracoelomal injection. The injection assays revealed pronounced differences in pathogenicity against the larvae.
Fusarium solani injection resulted in 100% lethality. Tests were conducted on the antibiotic sensitivity of suspected bacterial entomopathogens.
Thus, it can be concluded that the authors of the manuscript have received new data on the composition and structure of the microbiome in the intestines of the black soldier fly. In terms of larval health and yield, putative entomopathogenic agents of BSF were characterized and identified for the first time. Modern methods of statistical analysis were applied. References is quite complete, includes 83 sources. Excessive self-citation is not. The data obtained have both theoretical and practical importance. The presented manuscript can be published in the Microorganisms journal.
Author Response
-> We would like to thank reviewer #4 for the detailed report and the appreciation of our work.
Round 2
Reviewer 1 Report
Good job. I have no further comments. I am satisfied with the responses to the review received and the changes made.